# Efficacy and safety of upadacitinib in the treatment of moderate-to-severe atopic dermatitis: A systematic review

**Dan-Jie Zhao[1]☯, Xia Li[1]☯, Hai-Xia Lin[1]☯, Hong Zheng[1], Di Zhou[1], Peng Tang[2]\***

**1** Department of Pharmacology, The First People's Hospital of Shuangliu District, West China (Airport) Hospital of Sichuan University, Chengdu, Sichuan, China, **2** Department of TCM Pharmacy, Chengdu Integrated TCM and Western Medicine Hospital, Chengdu, Sichuan, China

☯ These authors contributed equally to this work.
\* tangpeng19940308@163.com

**Data Availability Statement:** All relevant data are within the paper and its Supporting Information files.

## Abstract

### Objective

To evaluate the efficacy and safety of upadacitinib in the treatment of moderate-to-severe atopic dermatitis (AD), and provide reference for rational clinical medication.

### Methods

PubMed, Medline, Embase, Web of Science, Clinical Trials Website, and Cochrane Library databases were searched from the time of establishment until January 6, 2024, to compile a list of all randomized controlled trials (RCTs) including upadacitinib in the treatment of moderate-to-severe AD. The quality of the included studies was evaluated using the Cochrane Systematic Review. Review Manager 5.3 software was utilized for statistical analysis of outcome measures.

### Results

A total of five studies were included in the meta-analysis. The results revealed that the 15 mg and 30 mg upadacitinib significantly improved Eczema Area and Severity Index (EASI) 75% {[Odds Ratio (OR) = 8.58, 95% confidence interval (CI) (5.84–12.60), $P < 0.00001$] [OR = 15.62, 95% CI (10.89–22.42), $P < 0.00001$]}, Numerical Rating Scale (NRS) $\geq$ 4 {[OR = 7.13, 95% CI (5.63–9.01), $P < 0.00001$] [OR = 11.30, 95% CI (8.93–14.31), $P < 0.00001$]}, and Investigator's Global Assessment (IGA) 0/1 {[OR = 8.63, 95% CI (6.60–11.27), $P < 0.00001$] [OR = 16.04, 95% CI (12.26–20.99), $P < 0.00001$]} compared to placebo. In terms of safety, although 15 mg and 30 mg upadacitinib significantly increased the overall adverse events rate compared to placebo {[OR = 1.31, 95% CI (1.09–1.58), $P = 0.004$] [OR = 1.85, 95% CI (1.54–2.21), $P < 0.00001$]}, there was no significant difference in the serious adverse events rate {[OR = 0.73, 95% CI (0.41–1.29), $P = 0.28$] [OR = 0.69, 95% CI (0.39–1.23), $P = 0.21$]} and withdrawal rate due to adverse events {[OR = 0.66, 95% CI (0.39–1.11), $P = 0.12$] [OR = 0.85, 95% CI (0.52–1.38), $P = 0.50$]} compared to placebo.

**Funding:** The author(s) received no specific funding for this work.

**Competing interests:** The authors have declared that no competing interests exist.

## Conclusion

This meta-analysis preliminarily suggests that upadacitinib is effective and safe for usage in the treatment of moderate-to-severe AD. Additionally, upadacitinib can instantly relieve itchiness and effectively reduce symptoms and signs, with its 30-mg dose being more effective than the 15-mg dose.

## 1. Introduction

Atopic dermatitis (AD) is a chronic inflammatory skin disease that affects approximately 11–20% of children and 5–8% of adults worldwide [1, 2]. Its incidence has been rising over the past decade, progressively elevating to the rank of one of the most debilitating skin diseases globally [3]. About 85–90% of patients experience symptoms during childhood, which frequently recur and may even worsen, severely affecting the patients' psychological well-being and quality of life. This poses significant challenges for the clinical treatment of patients with moderate-to-severe AD [4, 5]. Open ulcers on the skin surface that cause skin damage accompanied by intense and persistent itching and followed by dryness, cracking, pain, erythema or darkening, crusting, and exudation are among the symptoms of acute phase [6]. In the subacute phase, scaling and dry fissures manifest, and the skin eventually becomes lichenified [7].

Recent studies have reported that a single targeting is no longer the only method to treat AD, and multi-target therapy is the future direction [8]. Glucocorticoids, calcineurin inhibitors, immunosuppressants, phosphodiesterase-4 inhibitors, ion channel inhibitors, monoclonal antibodies, and Janus kinase (JAK) inhibitors are currently used as clinical drugs for treating AD [9–11]. Among these, JAK inhibitors are targeted small-molecule drug formulations with the advantages of immediate effectiveness and multi-target action. They have become a new therapy for treating AD, providing patients with moderate-to-severe AD with a new alternative for treatment [12].

Upadacitinib, a selective and reversible oral small-molecule drug that inhibits JAK1, has been approved by the European Medicines Agency and the U.S. Food and Drug Administration (FDA) for usage in patients with moderate-to-severe AD aged 12 years and above. The recommended dosage is 15 mg once daily, with the possibility to increase to 30 mg once daily in cases where the treatment response is deemed inadequate [13]. Its oral administration not only makes it convenient to carry and consume but also greatly improves patient compliance.

Although several studies have been published on the efficacy and safety of JAK inhibitors in treating AD [14–17], certain limitations exist. First, the safety and efficacy of upadacitinib as a monotherapy for AD have not been analyzed separately. Second, as a second-generation JAK inhibitor, upadacitinib differs from first-generation drugs such as tofacitinib in JAK selectivity. It remains elusive whether this difference directly impacts the efficacy and safety of the drug. Third, subgroup analysis based on upadacitinib dosage has not been conducted. Furthermore, since the FDA approved upadacitinib in 2022 for the treatment of moderate-to-severe AD, it has been extensively used in clinical practice. Multiple clinical trials [18–22] have reported that upadacitinib is not only highly effective in treating moderate-to-severe AD but also has a low risk of cancers, adverse cardiovascular events, and thrombosis. Therefore, the systematic assessment of the efficacy and safety of upadacitinib at varying doses in the treatment of AD holds both necessity and significance. Based on this situation, we aimed in this meta-analysis to evaluate the efficacy and safety of upadacitinib for the treatment of moderate-to-severe AD.

## 2. Methods

### 2.1 Ethical statements

No ethical approval is required because this is a literature-based study. This systematic review and meta-analysis was conducted in accordance with the PRISMA guidelines, our study has been registered in PROSPERO (CRD42024523199) [23].

### 2.2 Search strategy

PubMed, Medline, Embase, Web of Science, Clinical Trials Website and Cochrane Library databases were searched from the first record to January 6, 2024 using the following terms: "Atopic dermatitis" and "JAK inhibitors or Upadacitinib or ABT-494." Additional studies were retrieved by checking the reference lists of relevant studies. Only trials published in English were included.

### 2.3 Inclusion and exclusion criteria

Inclusion criteria: Design: Randomized Controlled Trials (RCTs); Population: Moderate-to-severe AD ≥ 1 year; Age: Age ≥ 12years old; Eczema Area and Severity Index (EASI) score ≥ 16; Investigator's Global Assessment (IGA) score ≥ 3; Atopic dermatitis involving ≥ 10% of the body surface area; Baseline weekly average of daily worst pruritus Numerical Rating Scale (NRS) ≥ 4.

Exclusion criteria: Patients that previously received JAK inhibitor drugs and other drugs to improve symptoms of AD patients were excluded. Reviews, conference abstracts, letters, retrospective or case series were excluded.

### 2.4 Interventions measures

According to the randomized controlled double-blind method, patients were divided into: ①
Experimental group: oral administration of upadacitinib 15 mg or 30 mg; ② Control group: oral administration of placebo with the same course and method as experimental group. Other intervention measures were consistent between the experimental group and the control group.

### 2.5 Outcome measures

Primary outcome: Percentage of participants achieving at least a 75% reduction in EASI from baseline at week 16 (EASI-75%); Percentage of participants achieving a reduction of ≥ 4 points from baseline in worst pruritus NRS at week 16 (NRS ≥ 4); Percentage of participants achieving IGA for AD of 0 or 1 with a reduction from baseline of ≥ 2 points at week 16 (IGA 0/1).

Secondary outcome: Percentage of participants achieving a 90% reduction from baseline in EASI score at week 16 (EASI-90%); Percentage of participants achieving a 100% reduction from baseline in EASI score at week 16 (EASI-100%); Percent change from baseline in EASI score at week 16 (EASI baseline score); Percent change from baseline in worst pruritus NRS at Week 16 (NRS baseline score); Percent change from baseline in Scoring Atopic Dermatitis (SCORAD) score at week 16 (SCORAD baseline score). The overall adverse events rate: During the course of the trial, the subjects had all physical abnormalities unrelated to the purpose of the treatment (AE); The serious adverse events rate: During the course of the trial, subjects were hospitalized for adverse events or events that were life threatening or resulted in permanent or significant damage to the body or organs serious adverse events (SAE); Withdrawal rate due to AE: During the course of the trial, subjects were withdrew from the study due to adverse events.

## 2.6 Data extraction and quality assessment

Two authors independently reviewed relevant studies to assess the accuracy of the retrieval process. Then they screened the titles and abstracts of the literature and, if necessary, reviewed the entire work. Any discrepancies were resolved through discussion with a third author. Authors, publication year, patient characteristics, interventions, number of cases, treatment duration, and outcome measures were among the extracted data. The quality of the literature included in this study was evaluated using the risk of bias assessment table provided by the Cochrane Handbook for Systematic Reviews.

## 2.7 Data analysis

Data analysis was conducted using Review Manager 5.3 (The Cochrane Collaboration, UK). Dichotomous variables (such as EASI-75%) and continuous variables (such as EASI baseline score) were expressed as odds ratios (ORs) and weighted mean differences (WMD) with their respective 95% confidence intervals (CI). The $\chi^2$ test was employed to assess the heterogeneity among the included studies. A fixed-effect model was applied when statistical heterogeneity was determined to be low ($P > 0.1$ and $I^2 \leq 50\%$). Conversely, if significant heterogeneity was present, a random-effects model was employed. Subgroup analysis and sensitivity analysis were conducted to investigate the sources of heterogeneity.

# 3. Results

## 3.1 Literature search results

Based on the search strategy, a total of 3,585 relevant articles were retrieved. Following the removal of 1,468 duplicate articles, a total of 2,117 articles were obtained. After carefully reviewing the titles and abstracts, 2,027 articles, including animal experiments, retrospective studies, and studies unrelated to the topic, were excluded. The remaining 90 pieces of literature were further screened through full-text reading, resulting in the exclusion of 69 reviews, five studies without Placebo control, four studies without reported relevant data, and seven open-label studies. Eventually, five RCTs including a total of 3,189 patients with moderate-to-severe AD, with 1,062 in the placebo group, 1,058 in the 15-mg upadacitinib group, and 1,069 in the 30-mg upadacitinib group were involved. Patients came from various regional and racial backgrounds, and the treatment duration was 16 weeks. The literature search and screening process are presented in Fig 1. The basic characteristics of the included studies are depicted in Table 1.

## 3.2 Quality assessment of included studies

This research incorporated a total of five studies [18–22], all of which were randomized controlled double-blind clinical trials. The authors provided detailed descriptions of their specific plans for random sequence generation and allocation concealment. They also reported any follow-up withdrawals and losses and demonstrated no risk of bias in terms of selection, implementation, measurement, loss to follow-up, and reporting. The overall quality of the clinical trials was high (show in Fig 2).

## 3.3 Results of meta-analyses

**3.3.1 EASI-75%.** All five studies [18–22] included in the analysis reported changes in the number of patients achieving EASI-75% following treatment with 15 mg and 30 mg upadacitinib. Since these studies demonstrated significant heterogeneity (15 mg, $P = 0.03$, $I^2 = 64\%$; 30 mg, $P = 0.06$, $I^2 = 55\%$), a random-effects model was used for analysis. The results indicated that the number of patients achieving EASI-75% was significantly higher in 15 mg group

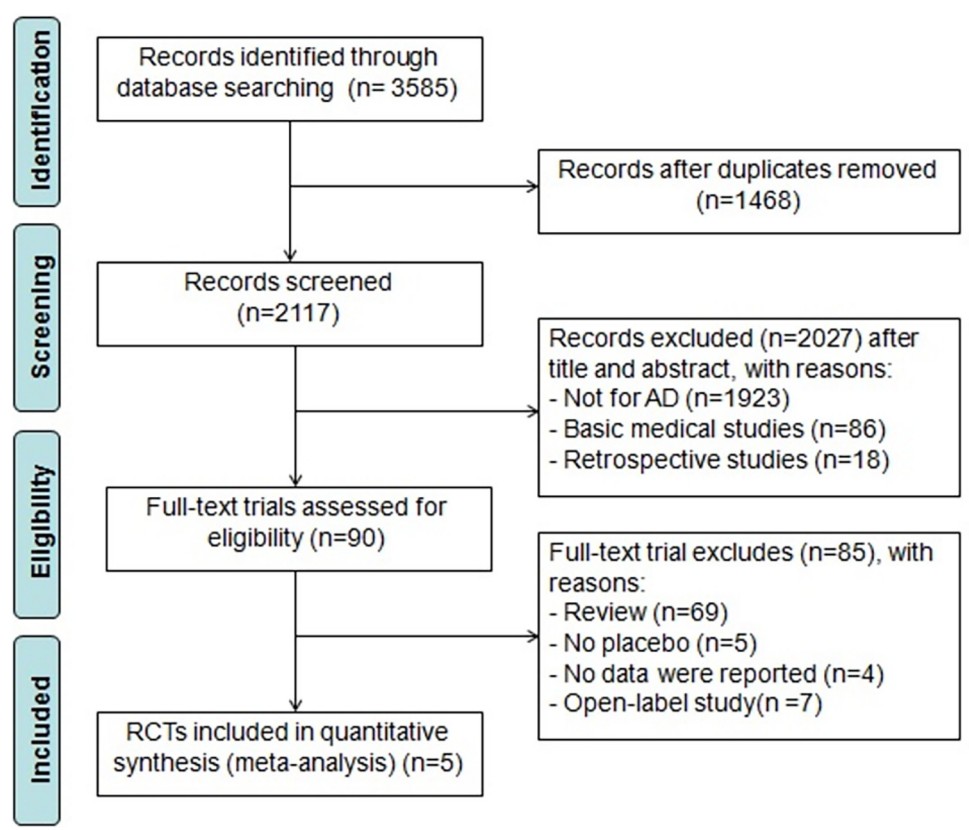

**Fig 1. The flow diagram of study selection.**

**Table 1. Study characteristics.**

| Study (Year) | Phase | Intervening measure | N | Age/Years | Gender/n | | Treatment/Week | Outcome |
|---|---|---|---|---|---|---|---|---|
| | | | | | **Female** | **Male** | | |
| Guttman YE (2020) [18] | Phase IIb | Upadacitinib 15 mg QD | 42 | 38.5±15.24 | 12 | 30 | 16 | ①②③④⑥⑦⑧⑨⑩⑪ |
| | | Upadacitinib 30 mg QD | 42 | 39.9±15.77 | 20 | 22 | 16 | |
| | | PBO | 41 | 39.9±17.52 | 17 | 24 | 16 | |
| Katoh N (2023) [19] | Phase III | Upadacitinib 15 mg QD | 91 | 35.9±13.2 | 23 | 68 | 16 | ①②③④⑨⑩⑪ |
| | | Upadacitinib 30 mg QD | 91 | 34.7±12.7 | 22 | 69 | 16 | |
| | | PBO | 90 | 36.3±12.6 | 16 | 74 | 16 | |
| Silverberg JI (2022) [20] | Phase III | Upadacitinib 15 mg QD | 321 | 31.3±12.02 | 131 | 190 | 16 | ①②③④⑤⑥⑦⑨⑩⑪ |
| | | Upadacitinib 30 mg QD | 320 | 34.0±13.40 | 120 | 200 | 16 | |
| | | PBO | 327 | 32.9±12.69 | 137 | 190 | 16 | |
| Simpson EL (2022) [21] | Phase III | Upadacitinib 15 mg QD | 301 | 31.7±13.65 | 140 | 161 | 16 | ①②③④⑤⑥⑦⑧⑨⑩⑪ |
| | | Upadacitinib 30 mg QD | 309 | 32.5±13.77 | 129 | 180 | 16 | |
| | | PBO | 302 | 31.9±12.64 | 139 | 163 | 16 | |
| Thyssen JP (2023) [22] | Phase III | Upadacitinib 15 mg QD | 303 | 32.7±14.80 | 137 | 166 | 16 | ①②③④⑤⑥⑦⑧⑨⑩⑪ |
| | | Upadacitinib 30 mg QD | 307 | 32.3±13.49 | 146 | 161 | 16 | |
| | | PBO | 302 | 33.1±12.94 | 147 | 155 | 16 | |

① EASI-75%; ② NRS ≥ 4; ③ IGA 0/1; ④ EASI-90%; ⑤ EASI-100%; ⑥ EASI baseline score; ⑦ NRS baseline score; ⑧ SCORAD baseline score; ⑨ AE; ⑩ SAE; ⑪ Withdrawal Rate due to AE.

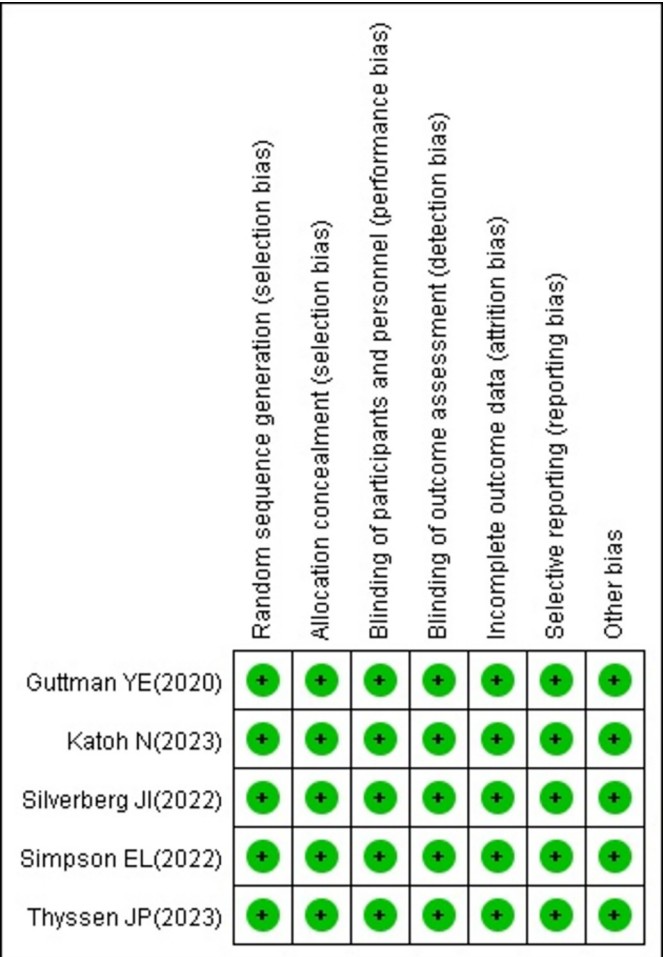

**Fig 2. Bias risk assessment chart.**

[OR = 8.58, 95% CI: 5.84–12.60] and 30 mg group [OR = 15.62, 95% CI: 10.89–22.42] compared to the placebo group, with both differences being statistically significant (P < 0.00001) (show in Fig 3).

**3.3.2 IGA 0/1.** All five studies [18–22] reported changes in the number of patients achieving IGA 0/1 following treatment with 15 mg and 30 mg upadacitinib. Given the minimal heterogeneity across the studies (15 mg, $P = 0.12$, $I^2 = 45\%$; 30 mg, $P = 0.38$, $I^2 = 5\%$), a fixed effects model was employed. The findings indicated a statistically significant increase in the number of patients achieving IGA 0/1 in 15 mg group [OR = 8.63, 95% CI: 6.60–11.27] and 30 mg group [OR = 16.04, 95% CI: 12.26–20.99] compared to the placebo group, with both differences demonstrating statistical significance ($P < 0.00001$) (show in Fig 4).

**3.3.3 NRS ≥ 4.** All five studies [18–22] reported changes in the number of patients achieving NRS ≥ 4 following treatment with 15 mg and 30 mg upadacitinib. Subsequently, minimal heterogeneity was observed among the studies (15 mg $P = 0.47$, $I^2 = 0\%$; 30 mg $P = 0.53$, $I^2 = 0\%$), prompting the utilization of a fixed effects model. The results revealed that the number of patients achieving NRS ≥ 4 was significantly higher in 15 mg group [OR = 7.13, 95% CI: 5.63–9.01] and 30 mg group [OR = 11.30, 95% CI: 8.93–14.31] compared to the placebo group, with both differences being statistically significant ($P < 0.00001$) (show in Fig 5).

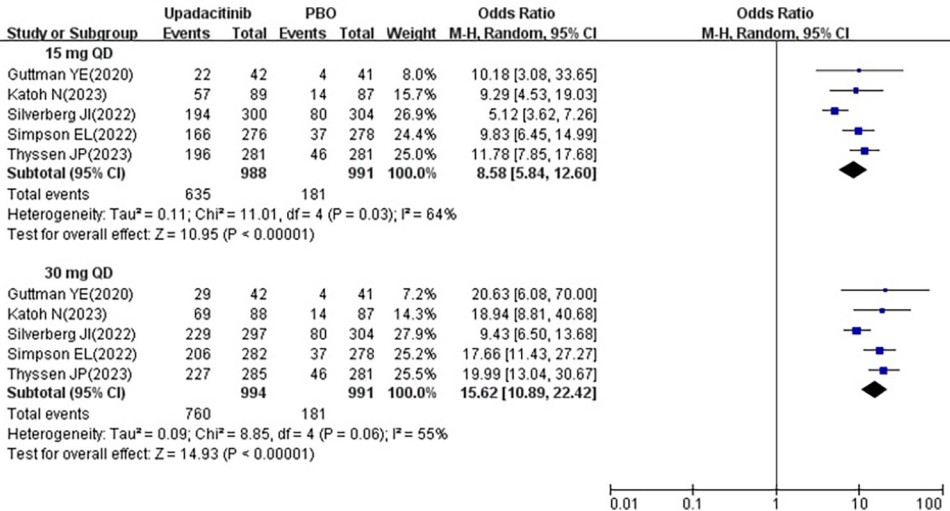

**Fig 3. Meta-analysis forest plot of EASI-75%.**

**3.3.4 AE.** All five studies [18–22] reported the AE following treatment with 15 mg and 30 mg upadacitinib. Minimal heterogeneity was observed among the studies (15 mg $P$ = 0.39, $I^2$ = 3%; 30 mg $P$ = 0.47, $I^2$ = 0%), leading to the adoption of a fixed effects model. The results indicated that the AE was higher in 15 mg group [OR = 1.31, 95% CI: 1.09–1.58] ($P$ = 0.004) and 30 mg group [OR = 1.85, 95% CI: 1.54–2.21] ($P$ < 0.00001) compared to the placebo group, with both differences being statistically significant (show in Fig 6).

**3.3.5 SAE.** All five studies [18–22] reported the SAE following treatment with 15 mg and 30 mg upadacitinib. Given the minimal heterogeneity among the studies (15 mg $P$ = 0.98, $I^2$ = 0%; 30 mg $P$ = 0.83, $I^2$ = 0%), a fixed effects model was employed. The results indicated that the SAE was no significant difference in 15 mg group [OR = 0.73, 95% CI: 0.41–1.29] ($P$ = 0.28) and 30 mg group [OR = 0.69, 95% CI: 0.39–1.23] ($P$ = 0.21) compared to the placebo group (show in Fig 7).

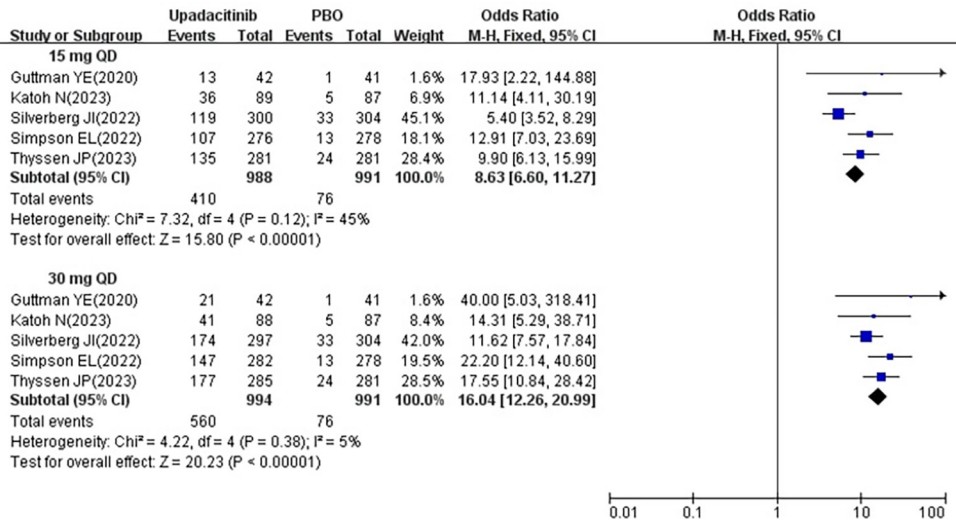

**Fig 4. Meta-analysis forest plot of IGA 0/1.**

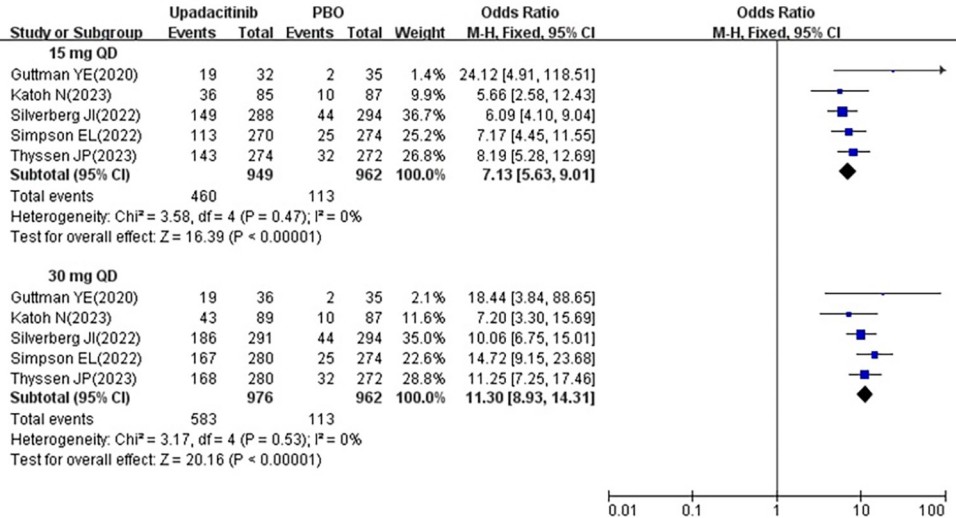

**Fig 5. Meta-analysis forest plot of NRS ≥ 4.**

**3.3.6 Withdrawal rate due to AE.** All five studies [18–22] reported the withdrawal rate due to AE following treatment with 15 mg and 30 mg upadacitinib. Given the minimal heterogeneity among the studies (15 mg $P = 0.57$, $I^2 = 0\%$; 30 mg $P = 0.64$, $I^2 = 0\%$), a fixed effects model was used. The results indicated that the withdrawal rate due to AE was no significant difference in 15 mg group [OR = 0.66, 95% CI: 0.39–1.11] ($P = 0.12$) and 30 mg group [OR = 0.85, 95% CI: 0.52–1.38] ($P = 0.50$) compared to the placebo group (show in Fig 8).

**3.3.7 Other efficacy measures.** We also conducted the meta-analysis on other efficacy indicators of AD. The results revealed that compared to the placebo, both 15 mg and 30 mg upadacitinib significantly increased the number of patients achieving EASI-90%, EASI-100% and 24 h NRS ≥ 4. Furthermore, they significantly reduced the EASI baseline score, NRS baseline score, and SCORAD baseline score, demonstrating significant efficacy in improving overall dermatitis area, itching severity and overall symptoms in patients, which was also dose-dependent (show in Table 2 and supplementary file).

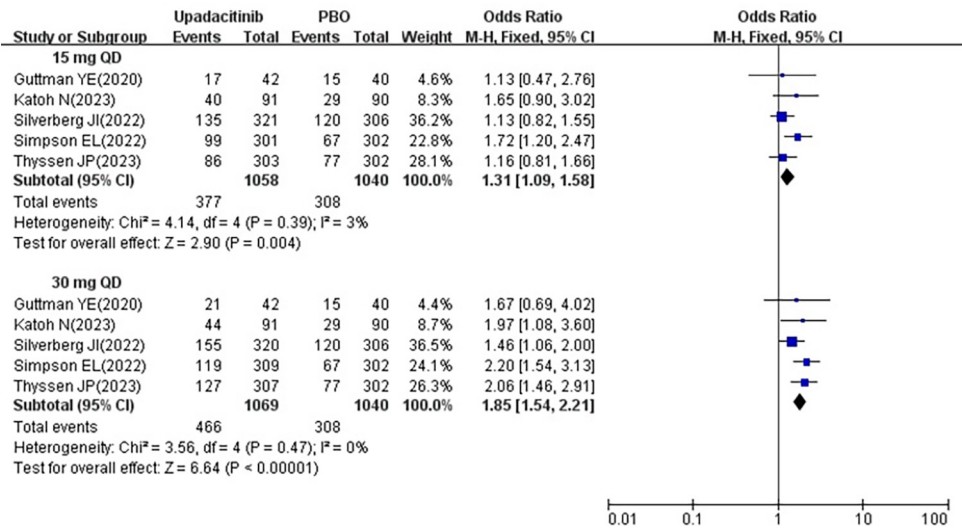

**Fig 6. Meta-analysis forest plot of AE.**

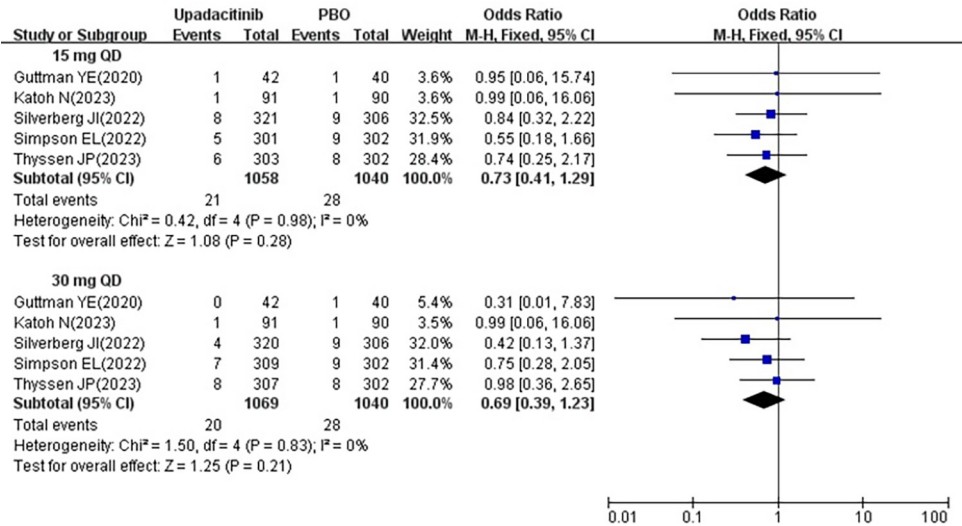

**Fig 7. Meta-analysis forest plot of SAE.**

**3.3.8 Other safety measures.** Due to the black box warning [24, 25], we conducted the meta-analysis on the specific occurrence rates of adverse reactions. The results revealed that compared to the placebo, both 15 mg and 30 mg upadacitinib significantly increased the risk of elevated blood creatine phosphokinase and acne formation. However, there was no significant difference in the occurrence rates of headache, Serious infection, and cancer. In addition, the occurrence rates of upper respiratory tract infection and nasopharyngitis were positively correlated with the dose. 30mg upadacitinib significantly increased the occurrence of upper respiratory tract infection and nasopharyngitis, while there was no significant difference between 15mg upadacitinib and placebo (show in Table 2 and supplementary file).

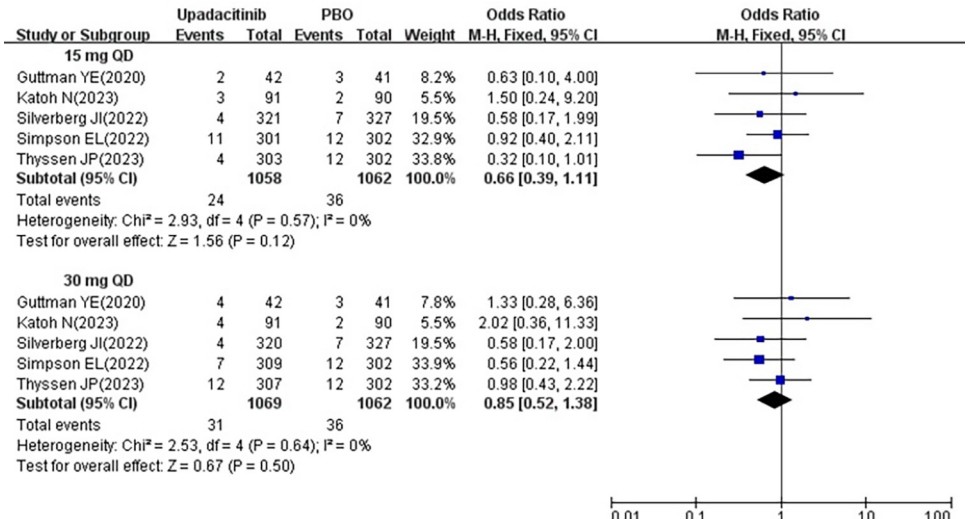

**Fig 8. Meta-analysis forest plot of withdrawal rate due to AE.**

**Table 2. Meta-analysis of other efficacy and other safety measures.**

| Outcome | Intervening measure | dosage | Study | $I^2$ | Analysis mode | WMD/OR | 95%CI | P value |
|---|---|---|---|---|---|---|---|---|
| EASI-90% | Upadacitinib vs PBO | 15mg | 5 [18–22] | 67% | Random-effect | 9.36 | (5.58, 15.69) | <0.00001 |
| | | 30mg | | 48% | | 17.24 | (11.47, 25.91) | <0.00001 |
| EASI-100% | Upadacitinib vs PBO | 15mg | 3 [20–22] | 0% | Fixed-effect | 12.89 | (6.90, 24.09) | <0.00001 |
| | | 30mg | | 0% | | 23.20 | (12.51, 43.02) | <0.00001 |
| 24h NRS ≥ 4 | Upadacitinib vs PBO | 15mg | 4 [19–22] | 41% | Fixed-effect | 11.56 | (6.09, 21.96) | <0.00001 |
| | | 30mg | | 36% | | 19.69 | (10.46, 37.08) | <0.00001 |
| EASI baseline score | Upadacitinib vs PBO | 15mg | 4 [18, 20–22] | 99% | Random-effect | -37.46 | (-41.91, -33.01) | <0.00001 |
| | | 30mg | | 99% | | -47.42 | (-51.98, -42.87) | <0.00001 |
| NRS baseline score | Upadacitinib vs PBO | 15mg | 4 [18, 20–22] | 77% | Random-effect | -35.02 | (-36.83, -33.20) | <0.00001 |
| | | 30mg | | 98% | | -48.90 | (-54.42, -43.38) | <0.00001 |
| SCORAD baseline score | Upadacitinib vs PBO | 15mg | 3 [18, 21–22] | 94% | Random-effect | -32.16 | (-35.09, -29.23) | <0.00001 |
| | | 30mg | | 93% | | -42.39 | (-45.22, -39.55) | <0.00001 |
| Blood Creatine Phosphokinase Increased | Upadacitinib vs PBO | 15mg | 5 [18–22] | 0% | Fixed-effect | 2.08 | (1.24, 3.47) | 0.005 |
| | | 30mg | | 0% | | 2.68 | (1.64, 4.40) | <0.0001 |
| Acne | Upadacitinib vs PBO | 15mg | 5 [18–22] | 0% | Fixed-effect | 4.23 | (2.75, 6.52) | <0.00001 |
| | | 30mg | | 0% | | 7.07 | (4.66, 10.74) | <0.00001 |
| Headache | Upadacitinib vs PBO | 15mg | 5 [18–22] | 0% | Fixed-effect | 1.35 | (0.90, 2.03) | **0.15** |
| | | 30mg | | 0% | | 1.43 | (0.96, 2.14) | **0.08** |
| Serious infection | Upadacitinib vs PBO | 15mg | 5 [18–22] | 0% | Fixed-effect | 1.01 | (0.38, 2.69) | **0.99** |
| | | 30mg | | 36% | | 0.62 | (0.20, 1.90) | **0.40** |
| Cancer | Upadacitinib vs PBO | 15mg | 5 [18–22] | — | Fixed-effect | 5.05 | (0.24, 105.63) | **0.30** |
| | | 30mg | | 0% | | 4.32 | (0.73, 25.59) | **0.11** |
| Allergic dermatitis | Upadacitinib vs PBO | 15mg | 5 [18–22] | 0% | Fixed-effect | 0.42 | (0.28, 0.63) | <0.0001 |
| | | 30mg | | 41% | | 0.16 | (0.09, 0.30) | <0.00001 |
| Upper respiratory tract infection | Upadacitinib vs PBO | 15mg | 5 [18–22] | 0% | Fixed-effect | 1.18 | (0.83, 1.68) | **0.36** |
| | | 30mg | | 0% | | 1.48 | (1.05, 2.08) | 0.02 |
| Nasopharyngitis | Upadacitinib vs PBO | 15mg | 5 [18–22] | 0% | Fixed-effect | 1.19 | (0.87, 1.64) | **0.28** |
| | | 30mg | | 0% | | 1.43 | (1.05, 1.95) | 0.02 |

### 3.4 Publication bias

According to Chapter 5 of the Cochrane Handbook for Systematic Reviews, a funnel plot is recommended to assess potential publication bias when the number of studies is at least 10. Owing to the limited number of studies included in the meta-analysis, the assessment of potential publication bias was not conducted.

## 4. Discussion

The pathogenesis of AD is a highly intricate condition primarily characterized by mutations in epidermal genes, impairment of skin barrier function, and immune dysregulation [26]. Ongoing research has discovered the significant involvement of the JAK/signal transducers and activators of transcription (STAT) pathway in AD-related immune responses [25]. This pathway, through the mediation of various key cytokines such as interleukin IL-4, IL-6, IL-13, and IL-15, among others, and its interaction with immune cells, keratinocytes, and peripheral sensory neurons, contributes to the propagation of inflammation and itchiness [27]. JAK inhibitors have emerged as prominent therapeutic targets for the treatment of AD, with notable members including JAK1, JAK2, JAK3, and tyrosine kinase 2 [28]. As a member of JAK1 inhibitors, upadacitinib can reduce the occurrence and development of AD by inhibiting the excessive

activation of JAK1. The mechanisms underlying its therapeutic effects are elucidated as follows: 1. Upadacitinib can inhibit the generation of pro-inflammatory factors such as IL-6, IL-15, interferon (IFN)-α, and IFN-γ, thereby reducing inflammation [29]. 2. Upadacitinib can promote the expression of neurotrophins and long-chain fatty acids, alleviating epidermal damage [30]. 3. By inhibiting the activation of STAT3, upadacitinib promotes keratinocyte differentiation and the expression of epidermal-related proteins and increases the production of filaggrin gene, loricin, and natural moisturizing factors. Ultimately, this reduces skin surface damage and ulceration [31, 32]. 4. Upadacitinib can reduce the proliferation of astrocytes in the spinal cord dorsal horn, thereby delaying or alleviating itching, especially chronic itching [33].

This study showed that after 16 weeks treatment with two different doses of upadacitinib. Patients demonstrated effective improvement in EASI baseline score, NRS baseline score, and SCORAD baseline score. Additionally, the number of patients achieving EASI-75%, EASI-90%, EASI-100%, IGA 0/1, and NRS ≥ 4 all significantly increased, with statistically significant differences ($P < 0.05$). Systematic analysis of patients experiencing 24 h NRS ≥ 4 within upadacitinib treatment demonstrated a significant reduction in itching symptoms in both dose groups. Subgroup analysis outcomes indicated that, in comparison to 15 mg upadacitinib, the 30 mg upadacitinib exhibited greater improvement, displaying dose-dependent characteristics. Therefore, upadacitinib can improve overall symptoms, signs, and eczema area in patients with moderate-to-severe AD, and rapidly alleviate itching. In terms of safety, the most common side effects of upadacitinib include acne, blood creatine phosphokinase increased, upper respiratory tract infection, nasopharyngitis and headache. The numbers of occurrences and total numbers of acne, blood creatine phosphokinase increased, upper respiratory tract infection, nasopharyngitis and headache with 15 mg upadacitinib vs 30 mg upadacitinib vs placebo were 105/1058 VS 166/1069 VS 27/1062; 45/1058 VS 58/1069 VS 22/1062; 71/1058 VS 88/1069 VS 61/1062; 91/1058 VS 108/1069 VS 78/1062; 57/1058 VS 61/1069 VS 43/1062, respectively. This data indicates that the incidence of adverse reactions was 5–10% except for acne, which was more than 10%. As a result of meta-analysis, the overall adverse events rate with upadacitinib was significantly higher than placebo ($P < 0.05$), and both doses of upadacitinib significantly increased the incidence of acne and elevated levels of blood creatine kinase ($P < 0.05$). Although the 15 mg upadacitinib did not yield significant differences in the incidence of nasopharyngitis and upper respiratory tract infections compared to the placebo, a significant difference was observed with the 30 mg upadacitinib. Therefore, to some extent, the occurrence of AE with upadacitinib appears to be dose-dependent. There were no significant differences in the SAE or withdrawal rate due to AE when compared to the placebo. Moreover, outcomes from subgroup analysis indicated that the 30 mg upadacitinib exhibited an increase in the SAE and withdrawal rate due to AE, albeit without statistically significant differences when compared to the placebo ($P > 0.05$). Overall, these findings suggest, to some extent, that the majority of adverse events induced by upadacitinib are mild and tolerable.

However, following the black box warning by the FDA regarding increased risks of cancer, thrombosis, severe cardiovascular events, and death associated with tofacitinib, upadacitinib, being a member of JAK inhibitors, was not exempted. Therefore, the conclusion drawn from the aforementioned safety research data, suggesting that adverse events are mild and tolerable, may not be entirely convincing. Hence, we summarized the adverse events mentioned in the black box warning: the numbers of occurrences and total numbers of cancer, thrombosis, severe cardiovascular events, and death with 15 mg upadacitinib vs 30 mg upadacitinib vs. placebo were 2/1058 vs 5/1069 vs 0/1062; 0/1058 vs 0/1069 vs 2/1062; 0/1058 vs 1/1069 vs 2/1062; 0/1058 vs 0/1069 vs 0/1062, respectively. This data indicates the absence of death associated with the use of upadacitinib, suggesting a favorable safety profile concerning thrombosis and

severe cardiovascular events. However, there may be an increased risk of cancer. Meta-analysis of the data demonstrated no significant difference in the cancer between upadacitinib and placebo ($P > 0.05$), demonstrating, to some extent, that upadacitinib has higher safety compared to first-generation JAK inhibitors. Furthermore, a study lasting up to 160 weeks proved that upadacitinib is overall safe and well-tolerated [34], adding a layer of assurance to its safety profile.

A growing body of clinical, epidemiological, and molecular evidence highlights variations in disease severity, duration, and age of onset across different ethnic populations affected by AD [35–38]. These differences, therefore, necessitate additional evaluations of drug reactions in patients with AD from different racial backgrounds in clinical trials, aiming to optimize treatment options [39]. The study, published last year, was the first to evaluate the differences in efficacy and safety of Upadacitinib among different ethnic populations [22]. It included three RCTs [20–22] and analyzed the variations in efficacy and safety after 16 weeks of Upadacitinib treatment in individuals from White, Asian, and Black/African American descent racial backgrounds. And the study concluded that there was no apparent racial diversity observed among different ethnic groups in terms of safety. Besides both White and Asian patients showed similar efficacy in terms of measures such as EASI-75%, IGA 0/1, and NRS $\geq$ 4. However, Black/African-American patients, although not achieving statistical significance, exhibited relatively lower levels of responsiveness compared to White and Asian patients. The reason for this poor responsiveness may be that erythema can be more difficult to detect in patients with darker skin types and this decreases the accuracy of effective index assessment among Black/African American patients to some extent [40]. Consequently, there is a need for additional racial-related clinical data for validation in the future.

This study included a total of five clinical trials [18–22], all of which were randomized, double-blind, multicenter high-quality studies. However, there were certain limitations. As a novel treatment method, upadacitinib necessitates additional RCTs, especially positive drug-controlled trials, to comprehensively evaluate its efficacy and safety. The available long-term clinical research data remains limited, comprising only one trial with a relatively small total sample size. Consequently, there is a need for additional data for verification in the future. Some outcome indicators exhibit significant heterogeneity, which may be linked to the variations in the duration of the disease among the included patients (three studies [20–22] included patients with a disease duration of more than 3 years).

## 5. Conclusions

In conclusion, this meta-analysis suggests that upadacitinib is superior to placebo in the treatment of moderate-to-severe AD, providing rapid relief of skin itching and effectively improving symptoms, signs, and quality of life. The 30 mg upadacitinib was more effective than the 15 mg upadacitinib, and adverse events were mild and tolerable. Moreover, the long-term safety profile was deemed acceptable. The risk of adverse events mentioned in the black box warning was low. Therefore, compared to other non-selective JAK inhibitors, upadacitinib may be a more convenient, effective, and safe choice for patients with moderate-to-severe AD.

## Supporting information

**S1 Checklist. PRISMA 2009 checklist.**
(DOC)

**S1 Fig. Meta-analysis forest plots of other efficacy measures.**
(DOC)

**S2 Fig. Meta-analysis forest plots of other safety measures.**
(DOC)

**S1 Data. Raw data.**
(XLSX)

## Author Contributions

**Conceptualization:** Dan-Jie Zhao, Peng Tang.

**Data curation:** Dan-Jie Zhao.

**Formal analysis:** Dan-Jie Zhao.

**Methodology:** Xia Li, Hai-Xia Lin, Peng Tang.

**Software:** Hong Zheng, Di Zhou.

**Writing – original draft:** Dan-Jie Zhao.

**Writing – review & editing:** Xia Li, Hai-Xia Lin, Peng Tang.

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
