## [Decision Letter · Decision Letter 0]

31 May 2024

PONE-D-24-11869Efficacy and Safety of Upadacitinib in the Treatment of Moderate-to-Severe Atopic Dermatitis: A Systematic ReviewPLOS ONE

Dear Dr. Tang,

Thank you for submitting your manuscript to PLOS ONE. After careful consideration, we feel that it has merit but does not fully meet PLOS ONE’s publication criteria as it currently stands. Therefore, we invite you to submit a revised version of the manuscript that addresses the points raised during the review process.

We look forward to receiving your revised manuscript.

Kind regards,

Feroze Kaliyadan, M.D.

Academic Editor

PLOS ONE

Journal Requirements:

Reviewers' comments:

Reviewer's Responses to Questions

**Comments to the Author**

1. Is the manuscript technically sound, and do the data support the conclusions?

Reviewer #1: Yes

Reviewer #2: Yes

Reviewer #3: Yes

2. Has the statistical analysis been performed appropriately and rigorously? 

Reviewer #1: Yes

Reviewer #2: Yes

Reviewer #3: N/A

3. Have the authors made all data underlying the findings in their manuscript fully available?

Reviewer #1: Yes

Reviewer #2: Yes

Reviewer #3: Yes

4. Is the manuscript presented in an intelligible fashion and written in standard English?

Reviewer #1: Yes

Reviewer #2: Yes

Reviewer #3: Yes

5. Review Comments to the Author

Reviewer #1: Good comprehensive article

My suggestion is to add frequency of adverse events .

Frequency of cancers and thrombotic events has been mentioned, please add frequency of more common side effects like headache and nasopharyngitis.

Reviewer #2: To mention in detail regarding the efficacy of upadacitinib with respect to variations in ethnicity. Is it equally effective in all races? If not how does it differ? Any difference in onset of getting a therapeutic response/ duration of treatment/ side effects?

Reviewer #3: Thank you for your work; the interest in using Upadacitinib is increasing to treat refractory and unrespondant atopic dermatitis to the (classical) therapeutic option. However, the safety profile is a crucial concern. This manuscript will provide an additional overview of using this medication.

6. PLOS authors have the option to publish the peer review history of their article (what does this mean?). If published, this will include your full peer review and any attached files.

Reviewer #1: No

Reviewer #2: No

Reviewer #3: No

---

## [Author Response · Author response to Decision Letter 0]

11 Jun 2024

Dear Editors and Reviewers:

Thank you for your letter and for the reviewers’ comments concerning our manuscript entitled “Efficacy and safety of upadacitinib in the treatment of moderate-to-severe atopic dermatitis: A systematic review”. Those comments are all valuable and very helpful for revising and improving our paper, as well as the important guiding significance to our researches. We have studied comments carefully and have made correction which we hope meet with approval. Revised portion are marked in the paper. The main corrections in the paper and the responds to the reviewers' comments are as follows:

Point by point reply to editors’and reviewers’ scomments:

Editor：1.Please ensure that your manuscript meets PLOS ONE's style requirements, including those for file naming. 

Response: Thank you for your careful suggestion. We have made revisions to the manuscript in accordance with the style requirements of PLOS ONE.

2. Please confirm at this time whether or not your submission contains all raw data required to replicate the results of your study. Authors must share the “minimal data set” for their submission. 

Response: Thank you for your thoughtful suggestion. The original data mentioned in the manuscript have been compiled into a file named "Raw data" and we are willing to share the dataset.

Response: Thank you for your careful suggestion. We have carefully checked each of these references to make sure they are complete and correct. In addition to this, we have added six references to support our argument for the efficacy and safety of upadacitinib with respect to variations in ethnicity. (references 35-40)

Reviewer #1: Good comprehensive article

My suggestion is to add frequency of adverse events .

Frequency of cancers and thrombotic events has been mentioned, please add frequency of more common side effects like headache and nasopharyngitis.

Response: Thank you for your affirmative and pertinent comments. According to your comments, we have added information about frequency of more common side effects in the discussion section, which has been marked in the paper. (lines 277-285)

Reviewer #2: To mention in detail regarding the efficacy of upadacitinib with respect to variations in ethnicity. Is it equally effective in all races? If not how does it differ? Any difference in onset of getting a therapeutic response/ duration of treatment/ side effects?

Response: Thank you for your valuable suggestion which we hadn't thought of. According to a literature review, we have launched a detailed description of the efficacy and safety of upadacitinib with respect to ethnic variations in the discussion section, which has been marked in the paper. Meanwhile, we have added six references to support our arguments. (lines 316-334; references 35-40)

Reviewer #3: Thank you for your work; the interest in using Upadacitinib is increasing to treat refractory and unrespondant atopic dermatitis to the (classical) therapeutic option. However, the safety profile is a crucial concern. This manuscript will provide an additional overview of using this medication.

Response: Thank you so much for your kind and encouraging comment! It would indeed be an honor for us if our research could be published in PLOS ONE.

We appreciate for Editors/Reviewers’ warm work earnestly, and hope that the correction will meet with approval.

Once again, thank you very much for your comments and suggestions.

Yours sincerely,

Peng Tang on behalf of the authors.

Corresponding author: Peng Tang at Department of TCM Pharmacy, Chengdu Integrated TCM and Western Medicine Hospital, Chengdu 610000, Sichuan, China. Email: tangpeng19940308@163.com.

---

## [Editor Report · Decision Letter 1]

19 Jun 2024

Efficacy and safety of upadacitinib in the treatment of moderate-to-severe atopic dermatitis: A systematic review

PONE-D-24-11869R1

Dear Dr. Tang,

We’re pleased to inform you that your manuscript has been judged scientifically suitable for publication and will be formally accepted for publication once it meets all outstanding technical requirements.

Kind regards,

Feroze Kaliyadan, M.D.

Academic Editor

PLOS ONE

---

## [Editor Report · Acceptance letter]

21 Jun 2024

PONE-D-24-11869R1 

PLOS ONE

Dear Dr. Tang, 

I'm pleased to inform you that your manuscript has been deemed suitable for publication in PLOS ONE. Congratulations! Your manuscript is now being handed over to our production team.

Kind regards, 

on behalf of

Dr. Feroze Kaliyadan 

Academic Editor

PLOS ONE